# Predicting diabetes mellitus metabolic goals and chronic complications transitions— analysis based on natural language processing and machine learning models

**Claudia C. Colmenares-Mejia**[1], **Andrés F. García-Suaza**[2], **Paul Rodríguez-Lesmes**[2*],
**Christian Lochmuller**[3], **Sara C. Atehortúa**[4], **J.E. Camacho-Cogollo**[3], **Juan P. Martínez**[2],
**Juliana Rincón**[1], **Yohan R. Céspedes**[1], **Esteban Morales-Mendoza**[1],
**Mario A. Isaza-Ruget**[1]

1 Fundación Universitaria Sanitas, Bogotá, Colombia, 2 Universidad del Rosario, Bogotá, Colombia,
3 Universidad EIA, Envigado, Colombia, 4 Universidad de Antioquia, Medellín, Colombia

* paul.rodriguez@urosario.edu.oc

## Abstract

### Objective

To estimate Diabetes mellitus (DM) progression at one and two years in terms of glycemic targets and development of complications.

### Research design and methods

We analyzed a retrospective cohort of adult DM patients treated in a Health Maintenance Organization in Colombia, including those with at least one glycosylated hemoglobin (HbA1c) measurement in 2018, 2019, and 2020. We defined four disease transition stages based on metabolic goals according to HbA1c levels and complications: 1. Within HbA1c goals and without complications; 2. Outside goals and without complications, 3. Within goals, but with complications, and 4. Outside goals and with complications. We applied Natural Language Processing (NLP) techniques to extract relevant clinical information from Electronic Health Records. Machine learning (ML) models were used to predict patient progression.

### Results

A total of 23,802 patients were included. Despite achieving initial glycemic control, more than 60% of patients who started within HbA1c targets and without complications developed chronic complications within two years. Our models, which achieved up to 80% accuracy and F1 scores above 74%, identified key predictors of disease progression. Adherence to dyslipidemia treatment guidelines significantly reduced the likelihood of HbA1c deterioration and complications, whereas non-adherence to pharmacological treatments increased the risk of complications. These findings suggest that HbA1c control alone is insufficient to prevent disease progression and that a more comprehensive

**Data availability statement:** All the code required to reuse the algorithms and replicate the results with the original data is available in the following repository: https://github.com/androdri1/diabetesprogression. The original data that support the findings of this study are available from EPS Sanitas. Restrictions apply to the availability of these data due to the involvement of sensible personal data (electronic health records), which were used under license for this study and under the authorization of the Unisanitas Ethics and Research Board. Any researcher that would like to get access to the dataset will be required to apply for access by discussing it with Unisanitas and EPS Sanitas research offices: investigaciones@unisanitas.edu.co.

**Funding:** This study was supported by the Ministry of Science, Technology and Innovation (Minciencias) of Colombia. Grant Number: 138-2021. Andrés García and Paul Rodríguez acknowledge support by Fulbright-Colciencias and Colombia Cientifica – Alianza EFI 60185 contract FP44842- 220-2018, funded by The World Bank through the Scientific Ecosystems, managed by the Colombian Ministry of Science, Technology and Innovation (MINCIENCIAS). The funders had no role in study design, data collection and analysis, decision to publish, or manuscript preparation.

**Competing interests:** The authors have declared that no competing interests exist.

management approach—including lipid control, kidney function monitoring, and improved adherence to clinical guidelines—is necessary.

## Conclusions

Patient compliance with pharmacological treatments, professional adherence to clinical practice guidelines, and lifestyle interventions play a crucial role in diabetes progression. While our models provide strong predictive capabilities, improving data quality and integration remains essential for better forecasting and intervention strategies.

## Introduction

Diabetes mellitus (DM) constitutes a chronic and long-term disease responsible for morbimortality worldwide; in 2019, it was reported to be accountable for 1.5 million direct deaths and 460,000 indirect, mainly through kidney disease complications [1]. Additionally, it is estimated that 534.6 million people were suffering from DM in 2021, and the number of people with this condition will rise to 783.2 million by 2045 [2]. DM significantly increases medical costs, causes loss of productivity, premature mortality, and other effects such as reduced quality of life [3]. For example, in 2019, diabetes-related healthcare expenditure was around USD 760 billion for adults aged 20–79 years worldwide, and it will be approximately USD 845 billion by 2045 [4].

Healthcare systems implement comprehensive care pathways through evidence-based Clinical Practice Guidelines (CPG), and adherence to these recommendations by patients and their families, physicians, and institutions is critical to reducing DM's short- and long-term complications. However, several barriers have been identified, such as health system management, lack of clarity and credibility of guidelines, knowledge of the health professionals, and patient's knowledge and sociocultural beliefs [5]. To tackle barriers from an institutional flank, it is possible to develop clinical tools that include structured information and unstructured data from Electronic Health Records (EHR) to support clinical decision-making to predict DM progression.

The literature usually develops predictive models using socio-demographic characteristics and biomarkers as standard predictors of diabetes-CKD progression [6–14]. These models typically consider only progression to CKD or development of some particular complication, but do not consider whether the patient is meeting HbA1c targets. Moreover, these models typically do not consider the health professional's adherence to CPG recommendations as an attribute, especially in recommendations on lifestyle changes and patients' attitudes and support networks. This is because, among other things, health professional and patient adherence is typically registered within the EHR's free text (instead of structured fields). The closest study, extracts information on past history of diseases in the context of a CKD progression model [15].

Our objectives are two. First, using Natural Language Processing (NLP) techniques and data from the EHR of a Health Maintenance Organization (HMO) in Colombia, we reached information not only on the patient's characteristics but also on their pharmacological compliance and the professional's adherence to the care pathways (non-pharmacological and pharmacological recommendations). Second, we develop machine learning (ML) models to estimate the progression of DM, in terms of metabolic goals and the development of complications, at one and two years and to determine critical variables in such transitions.

### Research design and methods

A retrospective cohort study was conducted. Population belongs to a Colombian HMO with extensive coverage in the national territory. EHR records of patients with a confirmed diagnosis

of DM and with at least one Glycosylated hemoglobin (HbA1c) measurement for 2018, 2019, and 2020 were included. We utilized ICD-10 codes registered in the EHR to identify patients with a diagnosis of diabetes mellitus type 1 or type 2, excluding gestational diabetes cases. Furthermore, to ensure accuracy, the status of these diagnoses was cross-checked by the auditing department of the Health Maintenance Organization (HMO). We received approval by UNISANITAS Ethics and Research Board (CEIFUS 2116–21; October 29, 2021). Consent was not obtained as the data was analyzed anonymously, and was provided by the insurance company on March 25, 2022.

## Transition model

We proposed a theoretical disease progression model with four stages based on metabolic goals defined by HbA1c levels (<6.5, <7.0, <7.5 according to patients' characteristics) suggested by the American Diabetes Association standards and the presence of chronic complications (retinopathy, cerebrovascular disease, and chronic kidney disease) [16]. According to the model, a patient with DM can be in four basic stages at the beginning of follow-up: 1. Within HbA1c goals without complications (ON-NOT), 2. Outside goals without complications (OUT-NOT), 3. Within goals, but with complications (ON-YES), and 4. Out of goals and with complications (OUT-YES). The patients can move from stages with no complications to stages with complications in a unidirectional way; that is, once they develop one of the complications, they are in the stage with complications and cannot return to a stage without complications. In contrast, the transitions between the stages on and off-goals are bidirectional since it is determined by their HbA1c levels, as indicated by the arrows in Fig 1. Likewise, a patient might remain in the same stage during the period.

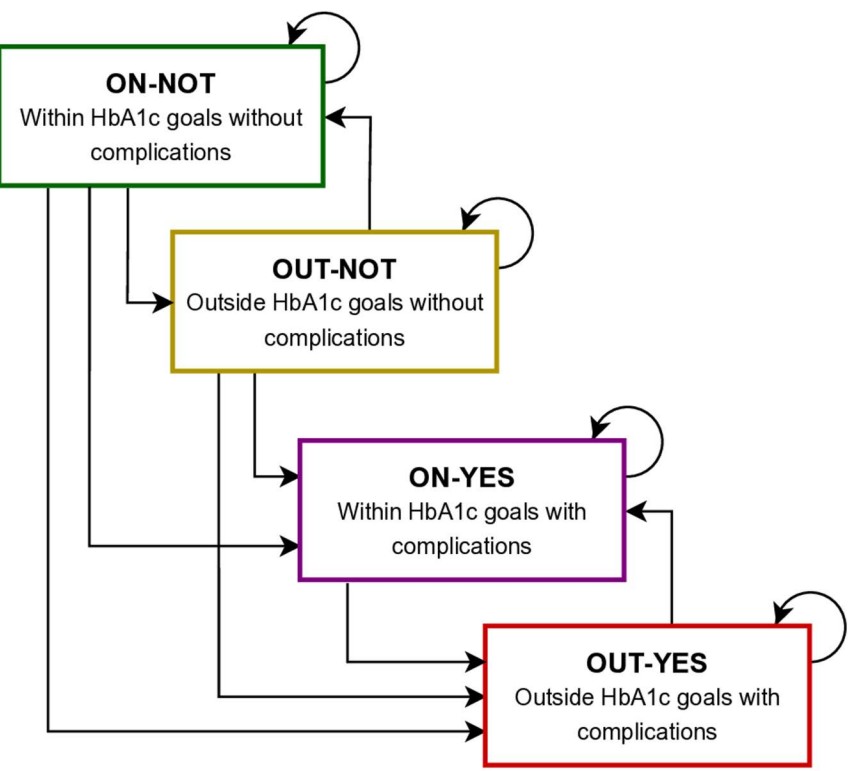

**Fig 1. Diagram of the transition model.**

## Data

The primary database comprised electronic health records (EHRs) that provided comprehensive information on outpatient care. This included demographic details such as age and sex, as well as complications like acute myocardial infarction, heart failure, peripheral vascular disease, chronic kidney disease, retinopathy, arrhythmias, and chronic obstructive pulmonary disease (COPD). Complications were documented using the International Statistical Classification of Diseases and Related Health Problems (ICD-10) (see Supplemental Material S1 File for the exact codes). It also covered measurements from physical examinations, including systolic and diastolic blood pressure, weight, height, and body mass index (BMI). Laboratory test results were part of the dataset, featuring HbA1c, LDL cholesterol, creatinine, and estimated glomerular filtration rate (eGFR) levels. The database also included information on diabetes mellitus treatment drugs, such as oral hypoglycemics and insulins, and concomitant treatments like analgesics, antacids, anticoagulants, antihypertensives, and lipid-lowering medications. Furthermore, the data encompassed referrals to specialists in ophthalmology, nutrition, psychology, and social services. It tracked professional adherence to personalized HbA1c goals, blood pressure goals, and dyslipidemia management. Additionally, free text information might provide insights into patient compliance with medication and adherence to medical recommendations for non-pharmacological interventions, including nutritional advice, physical exercise, and cessation of alcohol and tobacco use.

The information was recorded in a combination of standardized forms (structured data) and free-text fields (unstructured data), or both. Hence, before estimating transition models, we need first to consolidate a structured dataset. Table 1 presents the variables that were selected for the estimation of transition models due to their availability and relevance according to the CPG, and the source of information used to construct those variables.

## Deriving structured data from free text fields: the NLP pipeline

As an example of the goal of this data processing step, the Table 2 shows the information from the medical consultation of a patient. Originally, this patient had no data associated with complications or drugs in the EHR. However, free-text information has detailed information on both fields as well as information on non-adherence to drug therapy. Hence, we use NLP to classify the patient and to assemble the information on the missing fields: the patient has hypertension (bold and italic text) and is under several drugs that are typically used to treat diabetes or that are relevant for choosing a particular treatment strategy according to the guidelines (italic text). Notice that the words that indicate either health conditions or drugs may differ from the 'tags' that we use to identify the condition. For instance, the free text mentions "HBP" (high blood pressure) instead of using the word "hypertension". For the case of adherence to drug therapy (bold text), the procedure is more elaborated than searching for specific words, and the algorithm for assigning the indicator needs validation by a clinical expert.

We follow a three-step methodology to assign the corresponding labels to the unstructured data. We start by "cleaning" and "preparing" the data set (pre-processing). This involved removing URLs and special characters and digits, converting the text to lowercase, and removing stop words. Specific details of the pre-processing step are presented in supplemental material S2 File. Then, we extracted the relevant characteristics to classify the information (characterization). Next, we inferred the labels assigned to the records of interest (classification). This process creates new data, initially absent in the structured EHR fields, and then integrates it into the dataset.

**Table 1. Variables involved in the analysis.**

| Category | Origin | Variables | Clinical expert validation |
|---|---|---|---|
| ID | Structured data | Individual identification, Year | No |
| Demographics | Structured data | Age, sex | No |
| Complications | Structured data + ICD10* | Cerebrovascular disease, Chronic kidney disease, Retinopathy | No |
| Pharma. Treatments | Structured data + Simple search | Analgesics, antacids, Antihypertensives, hypoglycemic agents, Lipid-lowering agents | No |
| Referrals | Structured data + Simple search | Ophthalmology, Nutritionists, Psychology | No |
| Biomarkers | Structured data | LDL Cholesterol, eGFR, Creatinine, BMI, Weight, Height, Diastolic and Systolic Blood Pressure. | Yes |
| Adherence to CPG | Structured data + BoW | HbA1c guide adherence, cholesterol guide | Yes |
| Pharma. Adherence | BoW | Do patients follow pharmacological treatments as prescribed? | Yes |
| Recommendations | BoW | Nutrition, physical activity, alcohol and tobacco consumption | Yes |

Notes: *See Supplemental Material S1 for the exact codes. BoW: Bag-of-words.

**Table 2. Example of the derivation of structured data from free-text fields.**

| Patient id | Free text | Derived variables of complications and adherence | Derived variables of drugs |
|---|---|---|---|
| 81719 | 67 YEAR OLD PATIENT WITH **HBP** ON HANDLING WITH *INSULIN GLARGINE* 52 UNITS AT NIGHT ASPART 12-12-12 ASA 100X1 *OMEPRAZOLE* 20X1 *PREDNIS-OLONE* 5X1 *TESTOSTERONE* 250 MG MONTH *LOSARTAN+HYDROCHLOROTHIAZIDE* 50+12.5X2 *ROSUVASTATIN* 20X1 IMMUNOSUPPRESSANTS *CARBAMAZEPINE* 200X1 *CLOPI-DOGREL* 75X1 *MYCOPHENOLATE* 360MGX3 *METOPROLOL SUCCINATE* 50X1 *Empagliflozin+Metformin* 12.5mg/850mg Tab 1 e/12h | *Hypertension* | *Insulin glargine, Insulin Aspart, Omeprazole, Testosterone, Losartan + Hydrochlorothiazide, Rosuvastatin, Carbamazepine, Clopidogrel, Mycophenolate, Metoprolol Succinate, Emagliflozin+Metformin* |
| 10015 | Seventy-year-old patient with type II diabetes diagnostic. HbA1c on target, *hypertension is not controlled* due to a **decrease in the** *enalapril* **dose. Special emphasis on adhering to the pharmacological treatment** and on sticking to a healthy diet. General recommendations of a healthy lifestyle. Next appointment should be scheduled within 3 months. | *Hypertension* **Non-adherence to pharmacological treatment** | *Enalapril* |

**Note:** Example derived from a case in the dataset. Underlined: complications information. Italic: selected medications. Bold text: adherence to medication. Bold and italic: health conditions.

For the characterization and classification steps, we used two approaches to process the data: a simple search using a pipeline in which labels are generated from patterns in the free text and the Bag of Words (BoW) method [17,18]. The BoW method is ideal for scenarios where the simple search might not be straightforward (for instance, to detect if the physician considers that there is no pharmacological adherence) but where we can still rely on a pre-established "dictionary", a set of words that are relevant. This is the case as clinical literature uses relatively standard terms that indicate health conditions, treatments, and medications, among others. When such a dictionary is not available, NLP techniques should rely on

methods that guess the most relevant words from the free text (for instance the Term Frequency-Inverse Document Frequency TF-iDF method).

We created a custom medical dictionary to strengthen spelling correction, lemmatization, and stemming stages for text analysis. To construct it, we reviewed medical literature, including GPC and research articles, to ensure the vocabulary was complete and relevant to DM. Terms were carefully selected to include standard medical nomenclature, common variants, and colloquial terms that might appear in patient-reported data or EHR. The dictionary was validated by engineers and clinicians and tailored to the specific context of the DM. The Supplemental Material S2 File presents the details of how the BoW is implemented and the pre-processing steps involved.

With complete information on every field of interest in the database, we evaluated the level of adherence to the CPG recommendations. We assessed adherence in three key aspects: pharmacological, nutritional, and lifestyle recommendations such as tobacco and alcohol consumption. For medications, we considered whether the patients received adequate prescriptions for hypoglycemic agents, antihypertensives, and drugs to control cholesterolemia. Treatment inertia, the lack of a support, the attitude of the patient, or the incorrect prescription explain the discrepancies between the CPG algorithms and what is reported in the EHR. Again, we used NLP to examine these inconsistencies and have more precise patient adherence numbers.

## Estimation of transition models

Third, to estimate the predictive model of DM progression, we followed an architecture corresponding to a nested tree model in which one multiclass classification problem is transformed into two levels of binary classifications [19,20]. In particular, outcomes are defined as the stages in the model of Fig 1. Thus, we train two models per stage and time horizon, except for the stages ON-YES and OUT-YES, which only predict if patients will be in or out of goals. We had twelve different models since we predicted one and two years forward. Initially, we modeled the transitions in a multinomial setting, but the best results regarding the metrics described above were attained with the present modeling strategy. We compared the performance of several statistical models, including ML algorithms, as alternatives. In particular, we used K-nearest neighbors (KNN), logistic regression (LR), decision trees (DT), random forest (RF), neural networks (NN), and boosting (Boost).

The selection of ML models was guided by the characteristics of the dataset and the complexity of predicting diabetes progression. Specifically, LR was included due to its interpretability, making it a valuable tool for understanding the impact of individual predictors in binary classification tasks. Additionally, RF was chosen for its robustness to overfitting and ability to handle high-dimensional clinical data with complex interactions, which are prevalent in disease progression modeling. These models complement other approaches, such as KNN for benchmarking, DT for interpretability, NN for capturing nonlinear relationships and Boosting methods for handling imbalanced datasets. Supplemental material S3 File provides a detailed justification for selecting each model.

We profited from Pycaret (an ML library in Python) for automatizing part of the preprocessing tasks and for comparing performances across models. Pycaret allowed us to build the entire ML pipeline with minimal coding and to compare the performance across architectures.

For the preprocessing stage, we oversampled unbalanced classes and performed a feature selection that removed variables with low variance and that were highly collinear. In a nutshell, iteratively runs a LightGradientBoosting Machine model on different sets of characteristics to pick the most relevant ones for classification.

In the next step, the pipeline sets up a grid of models, where each model has a parameter grid, so we perform a grid search across both models and hyperparameters to determine which is the best predictor for a specific transition and outcome. We trained twenty models with a 10-fold cross-validation strategy (StratifiedKFold) and compared their performance in the validation set. This approach ensures that the model is trained on various data set partitions while maintaining the proportion of classes in each fold. In this case, cross-validation aimed to select hyperparameters and evaluate the model's generalization. Upon completion of the ten iterations, the model's performance at each fold is averaged to obtain an overall estimate of its performance on the data. This strategy ensures that the results reflect an average of several data set splits, providing a more robust metric. To determine the most adequate model in the gid search we considered as performances metrics the accuracy, the F1-score, and the Area Under the Curve (AUC). Among them, we privilege the F1-score. Considering the performance metrics, a common scenario when considering imbalanced datasets is that the F1 score exhibits proficiency, while the AUC appears suboptimal. The AUC can yield a lower value due to challenges in effectively distinguishing the minority class amidst the dominance of the majority class. Supplemental material 3 includes more details about the process and alternative strategies that we considered.

Notwithstanding Pycaret's ease of use, its performance with some models was quite low, so we turn to model a neural network with three fully connected layers, each one with 256 neurons and Rectified Linear Unit activation functions (ReLU). In between each layer we input a dropout layer that helps us regularize the weights and tackle overfitting. The final layer has a sigmoid activation function for modeling the transition probability. For each model we design a hyperparameter grid that varies the regularization parameter, the learning rate, and the epochs. Finally, we estimated marginal changes in the predicted transition probability to determine the relevance of variables included in the training.

## Results

### Population description, NLP results, and transitions

The original dataset started with 75,714 patients. Of those, only 25,320 had HbA1C measurements in the three study years, even though CPG suggest at least two measurements per year. Of those, a total 23,802 patients were included in the analysis as the remainder had missing information in some the predictor variables. These patients were dynamically distributed in the four stages (Table 3). The median age of the patients and the proportion of women in the four stages are similar. Some differences in biomarkers and results of physical examinations are explained by being at different stages. Interestingly, the proportion of compliant patients with pharmacological treatment is lower in the better scenarios (OUT-YES and ON-YES) than in the worst settings (ON-NOT and OUT-NOT). As expected, the proportion of patients treated by a professional who adhere to the HbA1c guideline is higher among those who are ON-goals than those who are out of goals. The proportion of patients receiving non-pharmacological recommendations is similar for patients ON-goals (independent of whether they have complications) and lower for those OUT-goals. In contrast, tobacco cessation recommendations seem more focused on patients with complications than patients without complications, regardless of their metabolic goals stage.

Using NLP, we found that 99% of patients received nutritional recommendations, 96% received advice on physical activity, 74% on alcohol consumption, and only 14% on tobacco use. In addition, NLP helps identify text patterns when analyzing professional adherence to the metabolic control guidelines algorithms. For example, Table 4 shows how the classification of patients receiving hypoglycemic treatment improved significantly after NLP (see differences between panels A and B in Table 4). Ideally, all patients would be on the diagonal of the matrix. However, in real clinical practice, patients may be reluctant to accept prescriptions, physicians may deviate from the CPG, or there may be treatment inertia instead of the CPG

**Table 3. Baseline characteristics by the transition stages, average for 2018-2020.**

| | ON-NOT | | | OUT-NOT | | | ON-YES | | | OUT-YES | | |
|---|---|---|---|---|---|---|---|---|---|---|---|---|
| | 2018 | 2019 | 2020 | 2018 | 2019 | 2020 | 2018 | 2019 | 2020 | 2018 | 2019 | 2020 |
| Number of patients starting in the stage | 8,685 | 5,376 | 3,127 | 6,858 | 3,860 | 2,027 | 4,375 | 8,202 | 10,874 | 3,884 | 6,362 | 7,774 |
| Characteristic/parameter[a] | Mean | SD[d] | | Mean | SD[d] | | Mean | SD[d] | | Mean | SD[d] | |
| Age (Years) | 69.34 | 11.95 | | 64.96 | 11.97 | | 70.41 | 11.75 | | 65.91 | 12.08 | |
| Female (proportion) | 0.6 | 0.49 | | 0.55 | 0.5 | | 0.59 | 0.49 | | 0.55 | 0.5 | |
| LDL Chol. (mg/dL) | 93.31 | 38.01 | | 96.33 | 40.88 | | 93.87 | 37.81 | | 96.56 | 39.73 | |
| eGFR (mg/g) | 76.83 | 20.23 | | 81.04 | 21.26 | | 77.55 | 20.23 | | 82.8 | 21.99 | |
| Creatinine (mg/dL) | 0.93 | 0.39 | | 0.91 | 0.39 | | 0.92 | 0.38 | | 0.9 | 0.4 | |
| BMI (Kg/m*m) | 28.84 | 4.79 | | 28.81 | 4.76 | | 28.8 | 4.79 | | 28.91 | 4.78 | |
| Weight (Kg) | 74.33 | 14.47 | | 75.3 | 14.79 | | 73.98 | 14.5 | | 75.2 | 14.57 | |
| Height (m) | 1.6 | 0.09 | | 1.61 | 0.09 | | 1.6 | 0.09 | | 1.61 | 0.1 | |
| Diast. B.P. (mmHg) | 75.29 | 7.64 | | 76.04 | 7.64 | | 75.23 | 7.51 | | 75.89 | 7.63 | |
| Sist. B.P. (mmHg) | 122.8 | 11.47 | | 123.5 | 12.03 | | 122.7 | 11.38 | | 123.4 | 11.93 | |
| HbA1c (%) | 6.37 | 0.49 | | 8.66 | 1.57 | | 6.43 | 0.46 | | 8.53 | 1.49 | |
| Patient compliance[b] | 0.12 | 0.33 | | 0.14 | 0.35 | | 0.15 | 0.36 | | 0.17 | 0.37 | |
| Cholesterol Adh.[b] | 0 | 0.07 | | 0.01 | 0.08 | | 0 | 0.06 | | 0.01 | 0.07 | |
| HbA1c guidelines adherence[b] | 0.73 | 0.44 | | 0.41 | 0.49 | | 0.64 | 0.48 | | 0.48 | 0.5 | |
| Physical act. recomm.[b] | 0.79 | 0.41 | | 0.81 | 0.39 | | 0.78 | 0.41 | | 0.82 | 0.39 | |
| Nutrition recomm.[b] | 0.85 | 0.35 | | 0.89 | 0.31 | | 0.85 | 0.36 | | 0.91 | 0.29 | |
| Tobacco cessation recomm.[b] | 0.07 | 0.26 | | 0.07 | 0.25 | | 0.05 | 0.23 | | 0.06 | 0.24 | |
| Alcohol cessation recomm.[b] | 0.45 | 0.5 | | 0.48 | 0.5 | | 0.42 | 0.49 | | 0.47 | 0.5 | |
| Antihypertensive treat.[c] | 1.11 | 1.14 | | 0.88 | 1.07 | | 1.17 | 1.15 | | 0.96 | 1.09 | |
| Hypoglycemic treat.[c] | 0.82 | 0.71 | | 1.3 | 1 | | 0.89 | 0.77 | | 1.49 | 1.06 | |
| Lipid-lowering treat.[c] | 0.01 | 0.11 | | 0.01 | 0.11 | | 0.01 | 0.12 | | 0.01 | 0.11 | |
| Analgesics[b] | 0.16 | 0.36 | | 0.1 | 0.3 | | 0.16 | 0.37 | | 0.11 | 0.32 | |
| Antacids[b] | 0.16 | 0.36 | | 0.13 | 0.34 | | 0.16 | 0.37 | | 0.14 | 0.34 | |

[a]Diast. B.P: diastolic blood pressure. eGFR: estimated glomerular filtration rate. Sist. B.P.: systolic blood pressure. Cholesterol Adh: professional adherence to dyslipidemia guideline. HbA1c guide Adh: professional adherence to glycosylated haemoglobin (HbA1c) guideline. recomm: recommendation. act: activity. treat: treatment. CKD: chronic kidney disease.

[b]proportion.

[c]number of drugs.

[d]SD, standard deviation.

recommendations. With NLP, we reduced the number of patients out of the diagonal due to the lack of structured data.

In general, for one-year transitions, patients tend to remain at their initial stage (see panel A in Table 5). However, at the two-year follow-up, patients without complications moved to other stages during the first year, while the transitions for patients with complications were stable (see panel B in Table 5). In particular, 18.1% of patients in the ON-YES stage moved out of goals after one year and 19% after two years. In addition, 26.9% of those who start in the OUT-YES stage tend to return to metabolic goals in the first year, and 32.9% do so in the second year, transitioning to the ON-YES stage. The second most common scenario for patients without complications and within goals is to develop some complications at one year (47.7% remain in ON-NOT, and 34.6% move to ON-YES). Analyzing two years, the scenario where the patients have complications becomes the most feasible: 52% move to the ON-YES stage, and only 26.8% remain in the ON-NOT stage. The group of patients initially in the OUT-NOT stage showed the most profound changes in transition patterns. In one year, 16.1% of

**Table 4. Patients' classification according to hypoglycemic drugs prescription: before and after NLP application.**

| A. Before NLP | | | | B. After NLP | | | |
|---|---|---|---|---|---|---|---|
| *Real*[b] | *Ideal*[a] | | | *Real*[b] | *Ideal*[a] | | |
| | *Metformin* | *Combined* | *Insulin* | | *Metformin* | *Combined* | *Insulin* |
| *Metformin* | 18.689 69,40% | 4.418 16,41% | 3.823 14,20% | *Metformin* | 18.686 74,16% | 4.418 17,53% | 2.093 8,31% |
| *Combined*[c] | 6.658 26,43% | 7.023 27,89% | 11.501 45,67% | *Combined*[c] | 6.658 33,90% | 7.023 35,76% | 5.961 30,35% |
| *Insulin* | 234 6,94% | 450 13,34% | 2.689 79,72%[d] | *Insulin* | 234 2,20% | 450 4,23% | 9.959 93,57%[d] |

[a]rows correspond to the ideal treatment for the patients.

[b]columns show what the patients were prescribed.

[c]Combined: metformin+second-line anti-diabetic.

[d]patients on insulin are correctly prescribed in 93.57% of the cases after the NLP procedure instead of 79.72% before applying the technique as example.

**Table 5. Transitions for patients with DM.**

| | Final stage | | | |
|---|---|---|---|---|
| | ON-NOT | OUT-NOT | ON-YES | OUT-YES |
| **Initial stage** | *A. 1-year transition* | | | |
| **ON-NOT** | 47.75% | 10.10% | 34.62% | 7.52% |
| **OUT-NOT** | 16.11% | 40.82% | 11.97% | 31.11% |
| **ON-YES** | 0.00% | 0.00% | 81.84% | 18.16% |
| **OUT-YES** | 0.00% | 0.00% | 26.94% | 73.06% |
| *B. 2-year transition* | | | | |
| **ON-NOT** | 26.86% | 6.73% | 52.04% | 14.37% |
| **OUT-NOT** | 11.57% | 21.03% | 22.33% | 45.06% |
| **ON-YES** | 0.00% | 0.00% | 80.99% | 19.01% |
| **OUT-YES** | 0.00% | 0.00% | 32.98% | 67.02% |

Notes: Panel A presents the analysis for patients who start without complications and panel B who start with complications. ON-NOT: within HbA1c goals and without complications, OUT-NOT: outside goals without complications, ON-YES: within goals, but with complications, and OUT-YES: out of goals and with complications.

them achieved goals, 40.8% remained in the same conditions, 11.9% only developed complications, and 31.1% moved to the worst scenario of OUT-YES. At two years, these percentages are 11.5%, 21%, 22.3%, and 45%, respectively.

## ML prediction models

Table 6 presents metrics for training and testing datasets for the set of the best algorithms according to their F1 score in the testing data, for each of the 12 models. Fig 2 illustrates the F1 scores for each of the models. Conditional to start in the better scenarios (ON-goals and NOT complications), the NN models have acceptable performance in predicting OUT-goals transition at one and two years (F1-scores between 0.72 and 0.76 in Table 6). For patients who start with the worst-case scenario (YES-complications), the LR fits well for predicting OUT-goals transition at one and two years (F1-scores > 0.8 for both). Predictions for those who remain in the same OUT-goals scenario were performed at one year with LR and two years with LGBoost, with good performance in both cases (F1-score of 0.83 and 0.78, respectively). The prediction of developing complications at one year was estimated using Boosting models for both ON-goals and OUT-goals starters (F1-score of 0.81 for AdaBoost and GBC). Similar

**Table 6. Performance of ML models of DM progression.**

|  | Initial condition | Prediction | Model | Training | | | Testing | | |
|---|---|---|---|---|---|---|---|---|---|
|  |  |  |  | Accuracy | F1-score | AUC | Accuracy | F1 | AUC |
| *1-year* | ON-goals | OUT-goals | NN | 0.9969 | 0.9969 | 0.9998 | 0.8083 | 0.7624 | 0.5805 |
|  | OUT-goals |  | LR | 0.7223 | 0.8388 | 0.5145 | 0.7216 | 0.8383 | 0.5499 |
|  | NOT-complications |  | NN | 0.9951 | 0.9951 | 0.9997 | 0.7783 | 0.7504 | 0.5745 |
|  | YES-complications |  | LR | 0.7358 | 0.8478 | 0.549 | 0.7355 | 0.8476 | 0.5258 |
|  | ON-goals | YES-complications | AdaBoost | 0.7153 | 0.8112 | 0.6296 | 0.6981 | 0.8011 | 0.6187 |
|  | OUT-goals |  | GBC | 0.722 | 0.8195 | 0.6275 | 0.6963 | 0.8011 | 0.5972 |
| *2-year* | ON-goals | OUT-goals | NN | 0.9944 | 0.9944 | 0.9999 | 0.7372 | 0.7263 | 0.6505 |
|  | OUT-goals |  | LGBoost | 0.6473 | 0.7804 | 0.5194 | 0.6407 | 0.7756 | 0.5178 |
|  | NOT-complications |  | NN | 0.9995 | 0.9995 | 1 | 0.7745 | 0.7409 | 0.5912 |
|  | YES-complications |  | LR | 0.6692 | 0.8016 | 0.5768 | 0.668 | 0.801 | 0.5067 |
|  | ON-goals | YES-complications | LR | 0.9031 | 0.9491 | 0.5236 | 0.9034 | 0.9492 | 0.5044 |
|  | OUT-goals |  | LR | 0.9073 | 0.9514 | 0.4901 | 0.9086 | 0.9521 | 0.5321 |

Notes: DM stands for Diabetes Mellitus, ON-NOT for a patient within HbA1c goals and without complications, OUT-NOT: outside goals without complications, ON-YES: within goals, but with complications, and OUT-YES: outside goals and with complications. Arrows indicate the stages that patients may transit over time. Complications refer to a diagnosis of retinopathy, cerebrovascular disease, or chronic kidney disease. NN: Neural network; LR: Logistic regression; AdaBoost: Adaptive Boosting; GBC: Gradient Boosting Classifier; LGBoost: Light gradient boosting.

predictions, but with superior performance, were made for the second year using LR (F1-score > 0.94 in both cases). The full set of results for all tested models is available in Supplemental Material S4 File. As discussed in the methods section, in imbalanced datasets, the F1 score gives favorable results when the predictive model excels in identifying the less prevalent but critical class (outside goals or with complications).

Finally, aiming to address the unbalanced nature of our training set, we tried two different adjustments to make all the classes equally representative. By means of the Synthetic Minority Oversampling Technique (SMOTE), (i) we upsampled the minority class, while holding the majority class constant; and (ii) we upsampled the minority class and downsampled the majority class. This was only performed on the training set while holding the test set constant. Even though there were minor gains in the cross-validation process, the testing metrics did not substantially change, as can be seen on Table 7 and Table 8. Given that the processing pipeline for the initial models was relatively simpler, we preferred to maintain them.

Table 9 presents the confusion matrix for the selected models. For its construction we selected a cutoff to translate predicted probabilities into binary predictions that maximize the kappa score in order to balance sensitivity (true positives rate) and specificity (true negatives rate). In general the resulting models perform better detecting true positives (predicting transitions that actually happened) than true negatives. In particular in the 2-year horizon, for those with complications the model has problems predicting that the patient will not move to OUT-goals, and the same for predictions into YES-complications from both ON and OUT-goals.

Aside from the performance metrics, from the general results from the model we can establish which factors exerted the most significant influence on predictions. Feature importance plots are provided for the models in rows two and four of Table 6, which correspond to Logistic Regressions predicting the one-year transition of HbA1c levels. Specifically, Fig 3 illustrates the most influential factors for patients without complications, while Fig 4 does so for patients with complications. For patients initially without complications, staying out of target ranges is associated with receiving nutrition recommendations and not adhering to clinical guidelines.

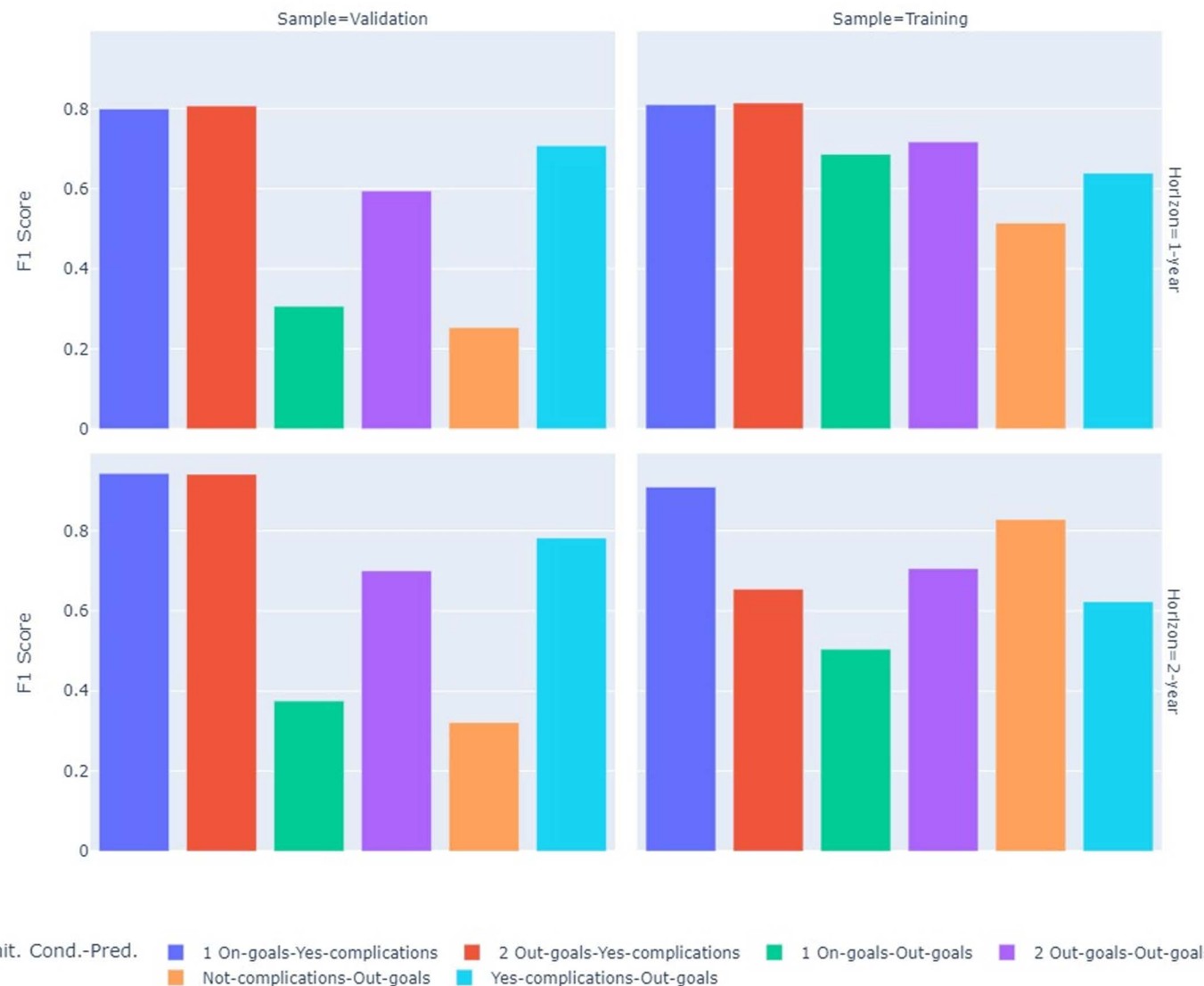

**Fig 2. Comparison of transitions models performance (F1 score).**

This underscores the critical role of prevention measures and proper nutrition in maintaining healthy HbA1c levels. Conversely, patients with early-stage Chronic Kidney Disease who do not follow the Cholesterol control guidelines also tend to miss their HbA1c targets. Overall, NLP-extracted features—particularly those related to nutritional habits—highlight the impact of dietary control in preventing HbA1c increases. Notably, lipid-lowering agents play a key role as well. Patients who consistently receive nutrition recommendations and take these medications often remain out of target, likely due to unchanged dietary habits.

We move from feature importance to understand the marginal effect of each variable on the transition probabilities. Figs 5 and 6 show predictors, such as age, explaining the probability of being OUT-goals at one and two years, regardless of whether the patient has complications. Fig 7 considers instead the role of predictors of developing complications at one year for patients OUT-goals, from which we highlight the importance of eGFR. This result is expected as most patients in this category develop CKD.

**Table 7.  Performance of ML models of DM progression after upsampling minority class while holding majority constant.**

|  | | | | Training | | | Testing | | |
|---|---|---|---|---|---|---|---|---|---|
|  | Initial condition | Prediction | Model | Accuracy | F1-score | AUC | Accuracy | F1 | AUC |
| *1-year* | ON-goals | OUT-goals | LGBM | 0.7621 | 0.7266 | 0.8045 | 0.799 | 0.0277 | 0.5348 |
|  | OUT-goals |  | DTC | 0.6446 | 0.6708 | 0.61 | 0.7053 | 0.8248 | 0.4988 |
|  | NOT-complications |  | ETC | 0.8151 | 0.7804 | 0.8559 | 0.7557 | 0.1828 | 0.5292 |
|  | YES-complications |  | LGBM | 0.7885 | 0.8399 | 0.8285 | 0.7065 | 0.8245 | 0.5208 |
|  | ON-goals | YES-complications | ETC | 0.8296 | 0.8301 | 0.895 | 0.5873 | 0.7043 | 0.5048 |
|  | OUT-goals |  | ETC | 0.7483 | 0.7524 | 0.8241 | 0.5894 | 0.7127 | 0.5015 |
| *2-year* | ON-goals | OUT-goals | ETC | 0.9219 | 0.9177 | 0.9747 | 0.7329 | 0.2115 | 0.5757 |
|  | OUT-goals |  | ETC | 0.8548 | 0.8571 | 0.9351 | 0.5951 | 0.7177 | 0.5248 |
|  | NOT-complications |  | LGBM | 0.8269 | 0.7458 | 0.8596 | 0.7511 | 0.1069 | 0.5825 |
|  | YES-complications |  | ETC | 0.6921 | 0.7621 | 0.7085 | 0.6033 | 0.7338 | 0.518 |
|  | ON-goals | YES-complications | ETC | 0.9605 | 0.9604 | 0.9952 | 0.8806 | 0.9362 | 0.5749 |
|  | OUT-goals |  | ETC | 0.9529 | 0.9515 | 0.9917 | 0.884 | 0.9382 | 0.5355 |

Notes: DM stands for Diabetes Mellitus, ON-NOT for a patient within HbA1c goals and without complications, OUT-NOT: outside goals without complications, ON-YES: within goals, but with complications, and OUT-YES: outside goals and with complications. Arrows indicate the stages that patients may transit over time. Complications refer to a diagnosis of retinopathy, cerebrovascular disease, or chronic kidney disease. LGBM: Light Gradient Boosting Machine; DTC: Decision Trees Classifier; ETC: Extra Trees Classifier.

**Table 8.  Performance of ML models of DM progression after upsampling minority class and downsampling majority.**

|  | | | | Training | | | Testing | | |
|---|---|---|---|---|---|---|---|---|---|
|  | Initial condition | Prediction | Model | Accuracy | F1-score | AUC | Accuracy | F1 | AUC |
| *1-year* | ON-goals | OUT-goals | ETC | 0.8292 | 0.8115 | 0.9011 | 0.7439 | 0.19 | 0.5502 |
|  | OUT-goals |  | LGBM | 0.7749 | 0.8325 | 0.8051 | 0.7053 | 0.8245 | 0.4926 |
|  | NOT-complications |  | LGBM | 0.8673 | 0.7768 | 0.9035 | 0.7937 | 0.068 | 0.5334 |
|  | YES-complications |  | LGBM | 0.7943 | 0.8444 | 0.8316 | 0.7191 | 0.8344 | 0.5236 |
|  | ON-goals | YES-complications | LGBM | 0.7126 | 0.7538 | 0.7718 | 0.6438 | 0.7441 | 0.6035 |
|  | OUT-goals |  | ETC | 0.7207 | 0.7938 | 0.7371 | 0.6473 | 0.7829 | 0.4955 |
| *2-year* | ON-goals | OUT-goals | ETC | 0.806 | 0.7101 | 0.8339 | 0.7177 | 0.1216 | 0.5132 |
|  | OUT-goals |  | ETC | 0.704 | 0.7667 | 0.7518 | 0.6025 | 0.7229 | 0.4998 |
|  | NOT-complications |  | LGBM | 0.8365 | 0.7736 | 0.8755 | 0.7521 | 0.1208 | 0.5906 |
|  | YES-complications |  | LGBM | 0.6945 | 0.7638 | 0.7256 | 0.6238 | 0.757 | 0.5088 |
|  | ON-goals | YES-complications | LGBM | 0.9206 | 0.9358 | 0.9637 | 0.8751 | 0.9334 | 0.4819 |
|  | OUT-goals |  | LGBM | 0.9224 | 0.9379 | 0.9708 | 0.8802 | 0.9359 | 0.4845 |

Notes: DM stands for Diabetes Mellitus, ON-NOT for a patient within HbA1c goals and without complications, OUT-NOT: outside goals without complications, ON-YES: within goals, but with complications, and OUT-YES: outside goals and with complications. Arrows indicate the stages that patients may transit over time. Complications refer to a diagnosis of retinopathy, cerebrovascular disease, or chronic kidney disease. LGBM: Light Gradient Boosting Machine; ETC: Extra Trees Classifier.

Professional adherence to HbA1c treatment recommendations reduces the probability (between 2 pp. and 5 pp. at one year and around 2 pp. at two years) of deviating from goals as long as the patients are on goals (Figs 5 and 6). Adherence to the dyslipidemia treatment guidelines decreases the probability of around 5 pp. of patients with complications remaining OUT-goals at two years (Fig 6). The probability of moving outside goals for patients in the better initial stage, i.e., ON-NOT, decreases more than 10% at one year (Fig 5) and only 1 pp. at two years (Fig 6) when the physicians adhere to the cholesterol recommendations. In addition, if professionals follow such guidelines, patients are less likely (20 pp. less for those ON-goals and around 3 pp. for OUT-goals) to develop complications at two years (Fig 7).

**Table 9. Confusion matrices.**

| Horizon | Initial condition | Prediction | Model | Training | | | | Validation | | | |
|---|---|---|---|---|---|---|---|---|---|---|---|
| | | | | TPR | FPR | TNR | FNR | TPR | FPR | TNR | FNR |
| 1-year | On-goals | Out-goals | NN | 0.669 | 0.0604 | 0.9396 | 0.331 | 0.5242 | 0.4087 | 0.5913 | 0.4758 |
| | Out-goals | | LR | 0.6932 | 0.6242 | 0.3758 | 0.3068 | 0.4853 | 0.3812 | 0.6188 | 0.5147 |
| | Not-complications | | NN | 0.5009 | 0.103 | 0.897 | 0.4991 | 0.23 | 0.1355 | 0.8645 | 0.77 |
| | Yes-complications | | LR | 0.5465 | 0.457 | 0.543 | 0.4535 | 0.665 | 0.5962 | 0.4038 | 0.335 |
| | On-goals | Yes-complications | AdaBoost | 0.9138 | 0.64 | 0.36 | 0.0862 | 0.9221 | 0.7207 | 0.2793 | 0.0779 |
| | Out-goals | | GBC | 0.8697 | 0.5331 | 0.4669 | 0.1303 | 0.9314 | 0.7506 | 0.2494 | 0.0686 |
| 2-year | On-goals | Out-goals | NN | 0.4908 | 0.1345 | 0.8655 | 0.5092 | 0.4737 | 0.309 | 0.691 | 0.5263 |
| | Out-goals | | LGBoost | 0.677 | 0.4763 | 0.5237 | 0.323 | 0.71 | 0.6287 | 0.3713 | 0.29 |
| | Not-complications | | NN | 0.8523 | 0.0514 | 0.9486 | 0.1477 | 0.5079 | 0.4168 | 0.5832 | 0.4921 |
| | Yes-complications | | LR | 0.5533 | 0.4507 | 0.5493 | 0.4467 | 0.9155 | 0.8602 | 0.1398 | 0.0845 |
| | On-goals | Yes-complications | LR | 0.9097 | 0.8462 | 0.1538 | 0.0903 | 0.9844 | 0.9551 | 0.0449 | 0.0156 |
| | Out-goals | | LR | 0.5067 | 0.4057 | 0.5943 | 0.4933 | 0.9728 | 0.9324 | 0.0676 | 0.0272 |

**Notes:** Results were computed by comparing the data with predictions derived after calculating a cutoff value per model that maximizes the kappa value. TPR: True positives rate (sensitivity). FPR: False positives rate. TNR: True negatives rate (specificity). FNR: False negatives rate. NN: Neural network; LR: Logistic regression; Ada-Boost: Adaptive Boosting; GBC: Gradient Boosting Classifier; LGBoost: Light gradient boosting.

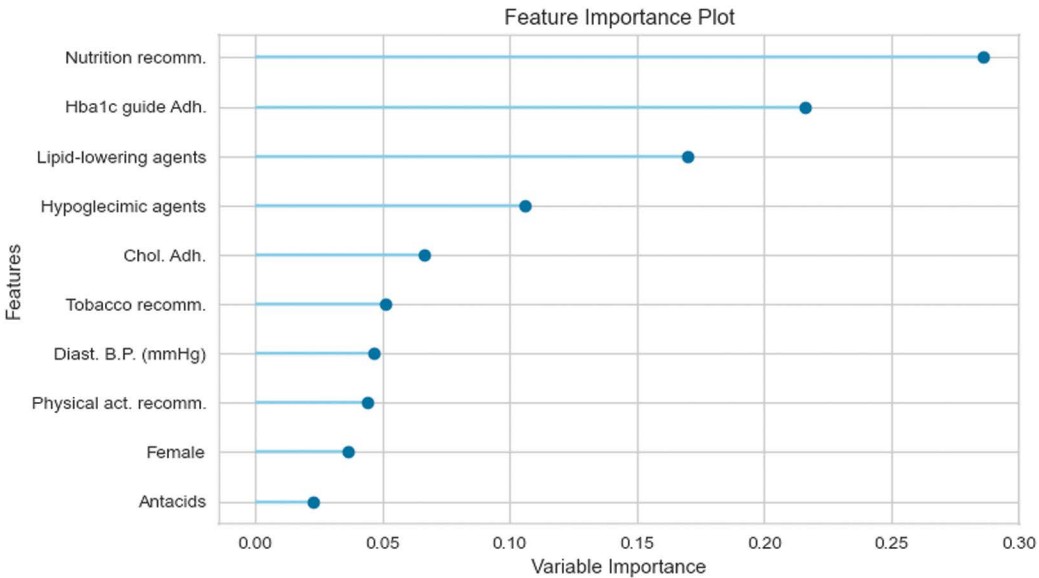

**Fig 3. Feature importance in predicting the probability of being off-target (conditional on having no complications and being off-target at t = 0).**

From the patients' side, non-compliance to pharmacological treatments increases, by more than 5 pp., the probability of developing complications at one year for the group of patients in the better scenario (see ON-NOT in Fig 7). Surprisingly, the direction of the effect of non-compliance on the probability of being out of the HbA1c goals at two years is the opposite for patients with complications compared with those without complications: the probability of being OUT-goals increases by 4 pp. if the patient belongs to the first group and decreases by around 3 pp. for the second (Fig 6).

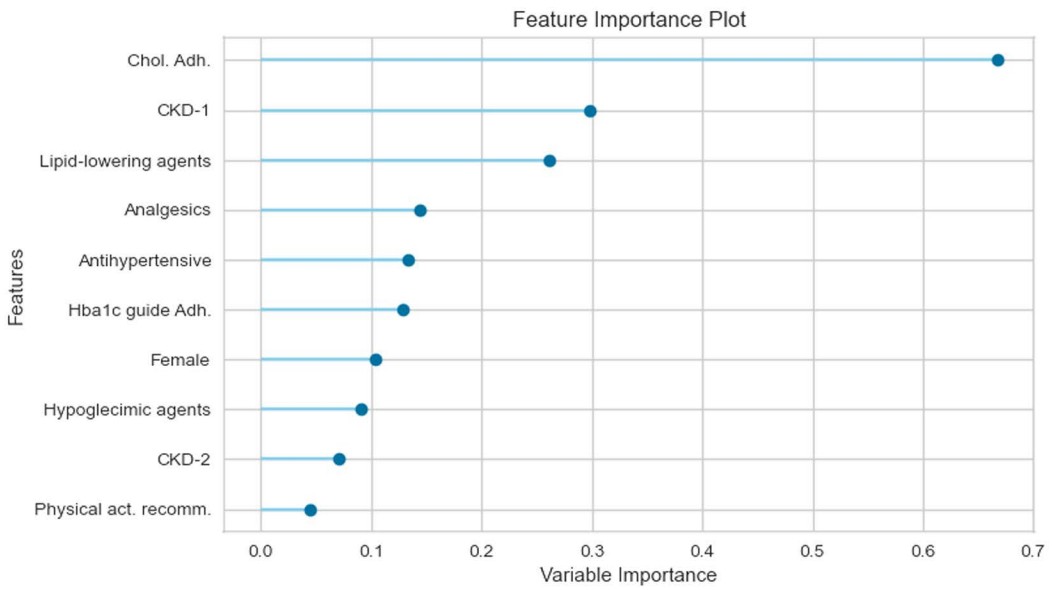

**Fig 4. Feature importance in predicting the probability of being off-target (conditional on having complications and being off-target at t = 0).**

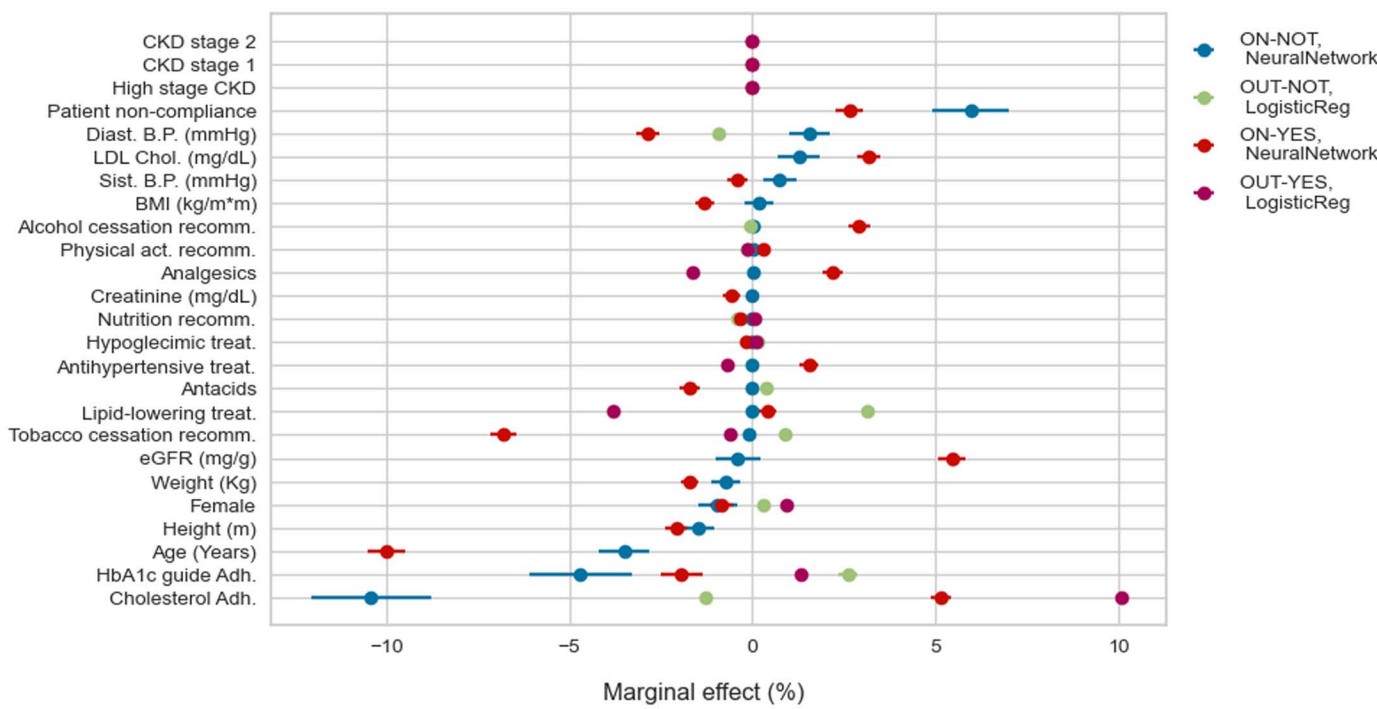

**Fig 5. Probability of moving outside of HbA1c goals in 1 year: marginal effects.**

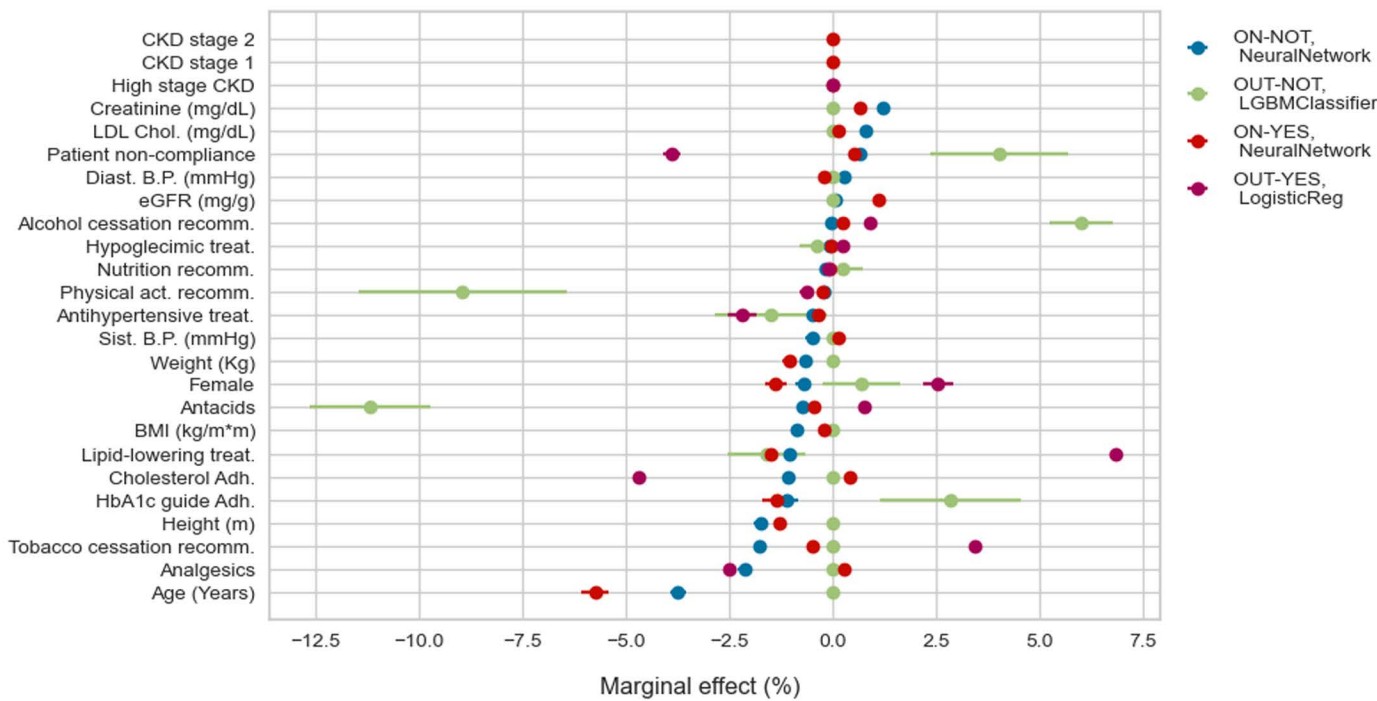

**Fig 6. Probability of moving outside of HbA1c goals in 2 years: marginal effects.**

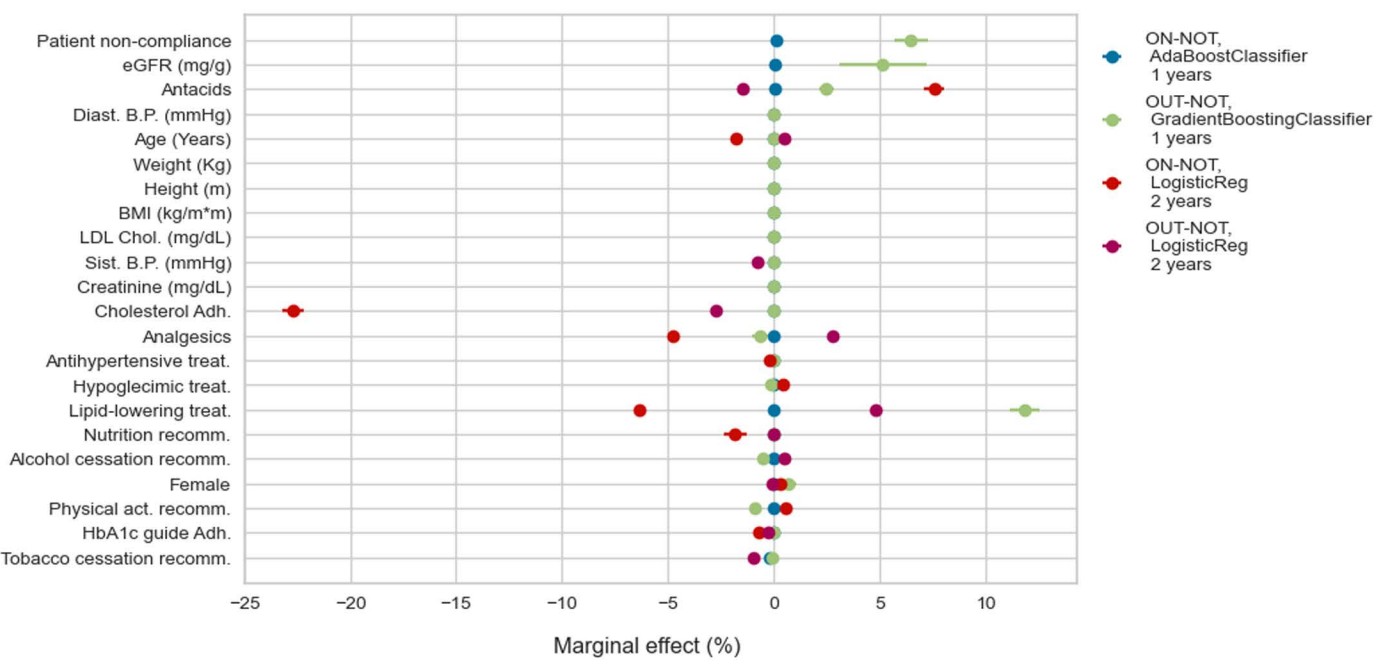

**Fig 7. Probability of developing complications: marginal effects.**

## Discussion

The transitions explored in the current analysis include a critical variable for the disease's management (metabolic goals) and dimensions involving multidimensional treatment factors. In particular, the model systematizes, in a replicable and scalable way, the information on the pharmacological compliance of the patient and the professional's adherence to the care pathways (non-pharmacological and pharmacological). In this sense, the model's predictions regarding the development of complications go beyond classic predictors such as age or creatinine levels, providing additional information on how institutional variables such as physicians' adherence to metabolic control guidelines and lifestyle recommendations play an essential role in the disease progression.

We run the models using a relatively similar number of patients and follow-up time compared to other studies in the literature [21]. Our models achieve close to 80% accuracy, similar to some results reported in the literature. For example, to predict the development of diabetes in China, the accuracy of RF models reaches up to 80% and NN 78% [22], and to predict risk factors for the progression of diabetic kidney disease to end-stage renal disease in the same Chinese context, RF models show accuracy around 82% [23]. In Japan, using big data machine learning methods with EHR data on a CKD aggravation model, the AUC was 0.743 and accuracy of 71% [15]. A CKD model, using data from a biobank, that on top of the EHR incorporated novel biomarkers for predicting the deterioration of renal function, reported an AUC of 0.77 [24]. Our models even outperform a recent Chinese study that used LR, DT, RF, and extreme gradient boosting (XGBoost) to predict progression to developing DM at 1 and 2 years (Accuracy around 60% and F1-scores close to 40%) [25].

This study could impact patients and the healthcare system from a clinical perspective. The model developed might aid in identifying individuals at risk of future complications early, enabling timely preventive measures. We highlight that the incidence of complications, despite achieving glycemic control, suggests the need for more comprehensive management strategies in patients with diabetes mellitus. For example, healthcare providers could implement more frequent monitoring of renal function (as indicated by the glomerular filtration rate) and cardiovascular health, especially in older patients or those with a history of poor compliance to treatment, even when they are within HbA1c targets. Additionally, our findings support the importance of reinforcing adherence to treatment guidelines for HbA1c and dyslipidemia, as well as personalized patient education focusing on lifestyle interventions, such as diet and exercise. These interventions could lead to better management of chronic complications like cardiovascular disease and kidney failure, ultimately improving long-term health outcomes in this population.

This analysis has several limitations. First, DM is a chronic disease usually related to complications incidence for five years, and our follow-up considers only 1 and 2 years, which is too short to identify all new cases in the cohort. Our model focuses on measuring the progression of DM towards critical stages like the development of complications. However, we had no data on more decisive outcomes such as hospitalization and mortality. Second, we recognize that the retrospective nature of our study and the inclusion criteria based on the availability of HbA1c measurements could introduce selection bias. Patients with more frequent follow-ups or better adherence to treatment might be overrepresented, potentially influencing the results. In addition, if the patients with diabetes who moved were more (or less) likely to develop complications or to be out of the sample, our selection would induce a bias. The usual transfer rate of general patients between HMOs in the country was around 6% [26], but we do not have such data for the HMO studied. Third, as our study was conducted within a single HMO in Colombia, the findings may not be directly generalizable to other populations or healthcare systems. Elements such as the NLP-generated labels are typically specific to the Colombian context, and to some extent to the

specific HMO. Still, the general procedure can be replicated elsewhere. For future work, we suggest validating the NLP exercise by obtaining metrics through the revision of the classifications by an independent clinical expert team in another dataset. In general, we recommend validating the presented prediction model in other populations prior to its clinical use.

In a more recent cohort of patients from the HMO studied, only 30% of patients with DM developed chronic complications, and about half were controlled. Surprisingly, we found that more than 60% of patients who start in the better scenario develop chronic complications at two years. Our result may be explained by characteristics specific to the study cohort analyzed. Considering the ML-model performance metrics, our F1 scores exhibit proficiency, while the AUC appears suboptimal and for some models the true negatives rate is not ideal. This phenomenon is observed in imbalanced datasets, where the instances of one class significantly outnumber those of the other [27].

## Conclusion

Using NLP techniques to incorporate unstructured information, we developed an ML-based model to estimate the probability of diabetes progression over one and two years in a cohort of 23,802 patients from a large Colombian HMO. Our findings reveal that glycemic control alone does not prevent disease progression. Despite starting in the best-case scenario—on-target HbA1c and no complications—more than 60% of patients develop chronic complications within two years. This highlights the need for a more comprehensive management approach beyond glucose control.

Our models also show that adherence to dyslipidemia treatment guidelines significantly reduces the likelihood of patients falling outside HbA1c targets and developing complications, while non-adherence to pharmacological treatments is a strong predictor of worsening outcomes. These results suggest that managing diabetes effectively requires enhanced monitoring of kidney and cardiovascular health, reinforcement of professional adherence to lipid and metabolic control guidelines, and improved patient education on lifestyle interventions.

Although our models perform well compared to existing literature, the results suggest that even with highly granular patient-level data from a major insurance company, predictions remain far from ideal. This underscores the importance of improving data collection and integration, particularly regarding lifestyle factors and long-term patient behaviors. Future work should focus on validating these findings in other populations and exploring interventions that can actively improve adherence and long-term outcomes.

The analysis here is based on correlations; in this sense, adherence to recommendations may be endogenous as doctors may be more incisive depending on what they expect their patients to do [28]. Further research should include some experimental variation, for instance, on the intensity of the exposure to the recommendations from the physicians. Finally, the ML models were packaged into a calculator that can be used as a decision-support tool in clinical practice. The calculator works like a simulator, allowing physicians to enter values of clinical variables and determine in real time the probability that the patient will move to different stages. Currently, the calculator is a test prototype designed and expected to be used for research purposes.

## Supporting information

**S1 File. List of ICD-10 codes used to characterize complications.**
(DOCX)

**S2 File. Details on the NLP algorithm.**
(DOCX)

**S3 File. Machine learning algorithm details.**
(DOCX)

**S4 File. All trained models' metrics.**
(CSV)

## Acknowledgments

We would like to thank Andrés Ramírez for the provided intellectual support and technical assistance during the elaboration of this paper, and Andrea Castillo Niuman for her contribution with administrative tasks. We also thank Fundación Universitaria Sanitas in Colombia for providing the primary dataset for this research.

## Author contributions

**Conceptualization:** Claudia C. Colmenares-Mejia, Andrés F. García-Suaza, Paul Rodríguez-Lesmes, Mario A. Isaza-Ruget.

**Data curation:** Juan P. Martínez, Yohan R. Céspedes, Esteban Morales-Mendoza.

**Formal analysis:** Claudia C. Colmenares-Mejia, Andrés F. García-Suaza, Paul Rodríguez-Lesmes, Sara C. Atehortúa.

**Funding acquisition:** Claudia C. Colmenares-Mejia, Andrés F. García-Suaza, J.E Camacho-Cogollo.

**Investigation:** Claudia C. Colmenares-Mejia, Andrés F. García-Suaza, Paul Rodríguez-Lesmes, Christian Lochmuller, Sara C. Atehortúa, J.E Camacho-Cogollo, Juan P. Martínez, Juliana Rincon, Yohan R. Céspedes, Esteban Morales-Mendoza, Mario A. Isaza-Ruget.

**Methodology:** J.E Camacho-Cogollo, Yohan R. Céspedes.

**Project administration:** Claudia C. Colmenares-Mejia, Andrés F. García-Suaza, J.E Camacho-Cogollo.

**Resources:** Claudia C. Colmenares-Mejia, Esteban Morales-Mendoza, Mario A. Isaza-Ruget.

**Supervision:** Andrés F. García-Suaza, J.E Camacho-Cogollo.

**Validation:** Claudia C. Colmenares-Mejia, Andrés F. García-Suaza, Paul Rodríguez-Lesmes, Christian Lochmuller, J.E Camacho-Cogollo, Juliana Rincon, Esteban Morales-Mendoza.

**Visualization:** Sara C. Atehortúa, Juan P. Martínez, Yohan R. Céspedes.

**Writing – original draft:** Sara C. Atehortúa.

**Writing – review & editing:** Claudia C. Colmenares-Mejia, Andrés F. García-Suaza, Paul Rodríguez-Lesmes.

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
