## [Decision Letter · Decision Letter 0]

13 Aug 2024

PONE-D-24-18044Predicting Diabetes Mellitus Metabolic Targets and Chronic Complications Transitions—Analysis Based on Natural Language Processing and Machine Learning ModelsPLOS ONE

Dear Dr. Rodríguez-Lesmes,

Thank you for submitting your manuscript to PLOS ONE. After careful consideration, we feel that it has merit but does not fully meet PLOS ONE’s publication criteria as it currently stands. Therefore, we invite you to submit a revised version of the manuscript that addresses the points raised during the review process.

We look forward to receiving your revised manuscript.

Kind regards,

Ozra Tabatabaei-Malazy

Academic Editor

PLOS ONE

Journal Requirements:

"This study was supported by the Ministry of Science, Technology and Innovation (Minciencias) of Colombia. Grant Number: 138-2021, Universidad del Rosario, Fundación Universitaria Sanitas, and Universidad EIA in Colombia. Andrés García and Paul Rodríguez acknowledge support by the Ministry of Science, Technology and Innovation (Minciencias) of Colombia. Grant Number: 138-2021"

"None"

Reviewers' comments:

Reviewer's Responses to Questions

**Comments to the Author**

1. Is the manuscript technically sound, and do the data support the conclusions?

Reviewer #1: No

Reviewer #2: Yes

Reviewer #3: Partly

2. Has the statistical analysis been performed appropriately and rigorously? 

Reviewer #1: N/A

Reviewer #2: Yes

Reviewer #3: No

3. Have the authors made all data underlying the findings in their manuscript fully available?

Reviewer #1: No

Reviewer #2: Yes

Reviewer #3: No

4. Is the manuscript presented in an intelligible fashion and written in standard English?

Reviewer #1: Yes

Reviewer #2: Yes

Reviewer #3: Yes

5. Review Comments to the Author

Reviewer #1: Summary:

In this study, the authors use natural language processing (NLP) and machine learning (ML) to predict diabetes progression from structured and unstructured electronic health records (EHRs).

Comments:

- The authors state that patients with a confirmed diagnosis of diabetes were included. Is confirmed diagnosis referred to ICD codes or specific diagnosis definitions used by the EHR system? Were only patients with type 2 diabetes included or also with type 1 diabetes?

- Why did the authors choose not to include neuropathy, peripheral vascular disease, and cardiovascular disease as complications?

- How were the complications defined? For chronic kidney disease, were also estimated glomerular filtration rates considered?

- The authors state that they used cross-validation to find the best performing model. However, they did not provide any information on whether cross-validation was just used for hyperparameter tuning or whether a nested cross-validation scheme was used to also get unbiased estimates of the models’ performance. Further, no information on the number of folds was provided.

- The results suggest that the ML models did not learn anything during training. For a binary classifier to achieve an F1-score of 0.95 and an AUC of 0.5, the dataset has to be very imbalanced (as is the case in the paper) and the predicted probabilities must be all very close to zero or one (depending whether the majority class labels are zero or one). The authors should carefully check their ML pipeline again.

- Mean HbA1c levels are not reported in Table 1.

- I would expect the authors to provide the full list of covariates that were used for the ML models.

- I would suggest removing Figure 1. It is clear from the text what is meant and the Figure does not look professional enough for a scientific publication.

- I am not an expert in NLP, thus I cannot assess whether the bag-of-words method is a reasonable choice for the task at hand.

Reviewer #2: This is a generally solid study in terms of both methodology and manuscript quality. The concept is interesting and somewhat unique within the broader landscape of published research. However, there are some areas where the manuscript could be improved, particularly by addressing certain missing details in the methodology:

Methods:

Population Description:

The description of the population lacks specifics regarding inclusion and exclusion criteria. For example, the phrase "patients with a confirmed diagnosis of DM" is vague. It would be helpful to specify the diagnostic criteria used for DM and clarify whether certain patient groups (e.g., those with a recent diagnosis, pregnancy, etc.) were excluded from the study.

NLP Methodology:

The Natural Language Processing (NLP) techniques described require more detailed explanation regarding the algorithms and parameters used. For instance, while you mention the creation of a custom medical dictionary, it is unclear how this was integrated into the NLP pipeline (e.g., which specific NLP models or tools, such as SpaCy, NLTK, etc., were utilized). Additionally, provide clarification on how the accuracy of the labels generated by the NLP methods was validated.

Model Performance Metrics:

The results section mentions F1 scores and accuracy for different models but lacks discussion on how these metrics compare across models and the reasons certain models outperformed others. Including a table summarizing the performance metrics for each model, along with a brief discussion of the strengths and weaknesses of each approach, would enhance the clarity of the findings.

ML Model Selection:

The manuscript references several machine learning algorithms (KNN, LR, DT, RF, NN, Boosting) but does not provide a rationale for selecting these specific models. Offering a brief justification for each model's selection based on the nature of the data and the research problem would strengthen the rigor of the methodology section. Additionally, more details on the hyperparameter optimization process and how it was tailored to each model would be beneficial.

Handling of Missing Data:

There is no discussion of how missing data were addressed, particularly in the EHR records. It would be helpful to specify the approach taken (e.g., imputation methods, exclusion of incomplete cases) and discuss the potential impact of missing data on the study’s results.

Discussion:

Clinical Implications:

The conclusion discusses the potential impact of the study but could be more specific. For instance, you mention the possibility of "improved health outcomes" without specifying which outcomes might be most affected. Consider providing concrete examples of how the findings could influence clinical practice, such as specific interventions that could be implemented This is a generally solid study in terms of both methodology and manuscript quality. The concept is interesting and somewhat unique within the broader landscape of published research. However, there are some areas where the manuscript could be improved, particularly by addressing certain missing details in the methodology:

Methods:

Population Description:

The description of the population lacks specifics regarding inclusion and exclusion criteria. For example, the phrase "patients with a confirmed diagnosis of DM" is vague. It would be helpful to specify the diagnostic criteria used for DM and clarify whether certain patient groups (e.g., those with a recent diagnosis, pregnancy, etc.) were excluded from the study.

NLP Methodology:

The Natural Language Processing (NLP) techniques described require more detailed explanation regarding the algorithms and parameters used. For instance, while you mention the creation of a custom medical dictionary, it is unclear how this was integrated into the NLP pipeline (e.g., which specific NLP models or tools, such as SpaCy, NLTK, etc., were utilized). Additionally, provide clarification on how the accuracy of the labels generated by the NLP methods was validated.

Model Performance Metrics:

The results section mentions F1 scores and accuracy for different models but lacks discussion on how these metrics compare across models and the reasons certain models outperformed others. Including a table summarizing the performance metrics for each model, along with a brief discussion of the strengths and weaknesses of each approach, would enhance the clarity of the findings.

ML Model Selection:

The manuscript references several machine learning algorithms (KNN, LR, DT, RF, NN, Boosting) but does not provide a rationale for selecting these specific models. Offering a brief justification for each model's selection based on the nature of the data and the research problem would strengthen the rigor of the methodology section. Additionally, more details on the hyperparameter optimization process and how it was tailored to each model would be beneficial.

Handling of Missing Data:

There is no discussion of how missing data were addressed, particularly in the EHR records. It would be helpful to specify the approach taken (e.g., imputation methods, exclusion of incomplete cases) and discuss the potential impact of missing data on the study’s results.

Discussion:

Clinical Implications:

The conclusion discusses the potential impact of the study but could be more specific. For instance, you mention the possibility of "improved health outcomes" without specifying which outcomes might be most affected. Consider providing concrete examples of how the findings could influence clinical practice, such as specific interventions that could be implemented based on the model's predictions.

Limitations:

The limitations section is somewhat underdeveloped. While you mention a short follow-up period and the lack of data on hospitalizations or mortality, other potential limitations, such as selection bias, the accuracy of NLP-generated labels, and the generalizability of the findings to other populations, are not addressed. Expanding on these aspects will provide a more comprehensive assessment of the study's limitations.

General:

Consistency in Terminology:

Ensure consistency in terminology throughout the manuscript. For example, the terms "out of targets" and "outside goals" are used interchangeably in different sections. Standardizing the language will help avoid confusion and improve readability.

Conclusion:

Incorporating more detailed explanations of the methodology and discussing the practical implications of the study in day-to-day clinical practice could significantly enhance the value of this already valuable study.

on the model's predictions.

Limitations:

The limitations section is somewhat underdeveloped. While you mention a short follow-up period and the lack of data on hospitalizations or mortality, other potential limitations, such as selection bias, the accuracy of NLP-generated labels, and the generalizability of the findings to other populations, are not addressed. Expanding on these aspects will provide a more comprehensive assessment of the study's limitations.

General:

Consistency in Terminology:

Ensure consistency in terminology throughout the manuscript. For example, the terms "out of targets" and "outside goals" are used interchangeably in different sections. Standardizing the language will help avoid confusion and improve readability.

Conclusion:

Incorporating more detailed explanations of the methodology and discussing the practical implications of the study in day-to-day clinical practice could significantly enhance the value of this already valuable study.

Reviewer #3: Dear Authors,

I hope this message finds you well.

We have reviewed your paper titled “Predicting Diabetes Mellitus Metabolic Targets and Chronic Complications Transitions—Analysis Based on Natural Language Processing and Machine Learning Models.” While your work provides a sophisticated insight into the application of hybrid machine learning methods in disease diagnosis, there are several major considerations that need to be addressed before further processing. Please review and address the following points:

Journal Style and Format: At first glance, the current context of your paper does not align with the typical structure of a journal article. It reads more like a summary report with limited data. Please ensure that your manuscript adheres to the journal’s style guidelines.

Literature Review: The literature review is insufficient. Typically, a robust review should include around 20 related articles focusing on the application of ML and NLP in disease diagnosis. Your review lacks depth and does not cover trends in disease complications. Please review at least 10 journal articles in your field that utilize machine learning and provide a comparative table of their results.

NLP Methods: You mention the implementation of NLP but do not specify which methods were used. There are three main NLP methods available. Please clarify which ones were implemented in your study.

NLP Metrics: You mentioned using two approaches, simple search and BoW, and ultimately selecting BoW. However, you did not report any clear metrics. If these metrics are in the appendix, please note that the appendix is not the appropriate place for them. Include clear metrics in the main text.

Machine Learning Methods: In the materials and methods section, you refer to both regression and classification methods, but your study focuses on classification. This makes the mention of regression methods unnecessary. It appears you implemented only neural networks but reported other methods as well. Please clarify which methods were implemented in sufficient detail.

Results and Discussion: For NLP, it is necessary to provide a table of metrics to compare all implemented groups, including data size (optional) and accuracy. Is this information in Table 2? Please clarify.

Classification Results: Clarify which classification method performed best and provide a consistent conclusion about the potential of your study, reducing unnecessary parts.

Figures: Please split Figure 2 into three separate figures, as it contains a lot of information. Combining them into one figure may reduce their quality.

Best regards

6. PLOS authors have the option to publish the peer review history of their article (what does this mean? ). If published, this will include your full peer review and any attached files.

**Do you want your identity to be public for this peer review?** For information about this choice, including consent withdrawal, please see our Privacy Policy .

Reviewer #1: No

Reviewer #2: **Yes: ** Sepehr Khosravi

Reviewer #3: **Yes: ** Dr. Parviz Narimani

---

## [Author Response · Author response to Decision Letter 1]

27 Sep 2024

Bogotá, September 11th 2024

Dear Editor and reviewers of PLoS ONE

We thank you for your time and effort in reviewing our manuscript entitled "Predicting Diabetes Mellitus Metabolic Targets and Chronic Complications Transitions—Analysis Based on Natural Language Processing and Machine Learning Models.” Detailed responses to reviewer’s comments are given below. In the manuscript, changes in content have been highlighted in red inside the main document.

Kind regards,

The authors

---

## [Decision Letter · Decision Letter 1]

22 Oct 2024

PONE-D-24-18044R1Predicting Diabetes Mellitus Metabolic Goals and Chronic Complications Transitions—Analysis Based on Natural Language Processing and Machine Learning Models.PLOS ONE

Dear Dr. Rodríguez-Lesmes,

Thank you for submitting your manuscript to PLOS ONE. After careful consideration, we feel that it has merit but does not fully meet PLOS ONE’s publication criteria as it currently stands. Therefore, we invite you to submit a revised version of the manuscript that addresses the points raised during the review process.

We look forward to receiving your revised manuscript.

Kind regards,

Ozra Tabatabaei-Malazy

Academic Editor

PLOS ONE

Journal Requirements:

Reviewers' comments:

Reviewer's Responses to Questions

**Comments to the Author**

1. If the authors have adequately addressed your comments raised in a previous round of review and you feel that this manuscript is now acceptable for publication, you may indicate that here to bypass the “Comments to the Author” section, enter your conflict of interest statement in the “Confidential to Editor” section, and submit your "Accept" recommendation.

Reviewer #2: All comments have been addressed

Reviewer #4: All comments have been addressed

2. Is the manuscript technically sound, and do the data support the conclusions?

Reviewer #2: Yes

Reviewer #4: Yes

3. Has the statistical analysis been performed appropriately and rigorously? 

Reviewer #2: Yes

Reviewer #4: Yes

4. Have the authors made all data underlying the findings in their manuscript fully available?

Reviewer #2: Yes

Reviewer #4: Yes

5. Is the manuscript presented in an intelligible fashion and written in standard English?

Reviewer #2: Yes

Reviewer #4: Yes

6. Review Comments to the Author

Reviewer #2: (No Response)

Reviewer #4: Dear Authors,Thanks for implementation of previous comments on the paper;moreover, with the minor revision on these areas, the paper would be acceptabe:

Detailed NLP Preprocessing Steps:

Clarity and Reproducibility: The manuscript would benefit from a more detailed description of the NLP preprocessing steps. For instance, specifying techniques used for text normalization (e.g., stemming, lemmatization), handling of stopwords, and dealing with medical abbreviations and jargon could improve clarity. Explaining these choices can help other researchers replicate your findings and understand the basis of your NLP pipeline’s effectiveness.

Error Analysis: Discuss any challenges faced during the text preprocessing, such as misclassifications or unusual patterns in the data. Addressing how these were handled or mitigated would provide insights into the robustness of the NLP methodology.

Justification for ML Algorithm Selection:

Algorithm Suitability: While the manuscript employs a variety of ML algorithms, there is little discussion on why these specific algorithms were chosen over others. For example, detailing the reasons for selecting neural networks or decision trees based on their suitability for the dataset characteristics (e.g., size, feature type, non-linearity) would enrich the analysis.

Performance Comparison: Provide a comparative analysis of the ML algorithms used. This could include discussions on each algorithm's performance with respect to handling imbalanced data, computational efficiency, and ease of interpretation. Such comparisons could be instrumental in justifying the choice of one model over another.

Model Validation Techniques:

Validation Strategy: The manuscript mentions the use of cross-validation but does not specify the type (e.g., k-fold, leave-one-out). Detailing the choice of cross-validation method, the number of folds used, and how the data was split can affect model evaluation significantly, especially in medical data where patient independence is crucial.

Handling of Imbalanced Data: Medical datasets often have class imbalance issues, particularly in disease prediction. Discussing strategies employed to handle imbalance, such as synthetic minority oversampling technique (SMOTE) or adjusting class weights in model training, would be valuable.

Hyperparameter Optimization:

Optimization Details: Expanding on the hyperparameter optimization process would strengthen the methodological rigor. For instance, detailing the parameters explored, the range of values tested, and the optimization algorithm (e.g., grid search, random search, Bayesian optimization) could provide deeper insights.

Impact on Model Performance: Discuss how hyperparameter tuning affected the performance of each model. This could include changes in model accuracy, sensitivity, specificity, and computational demands. Such details could help justify the final model configuration chosen for the study.

Performance Metrics Explanation:

Metric Selection Rationale: While accuracy, F1 score, and AUC are standard, discussing why these were chosen and their relevance to the study’s clinical implications would be insightful. For instance, in a medical context, the cost of false negatives might be more critical than false positives.

Thresholds and Trade-offs: Elaborate on any thresholds used for classification decisions (e.g., cut-off probabilities for classifying a patient’s disease stage). Discussing the trade-offs between sensitivity and specificity in choosing these thresholds, especially how they might impact clinical decisions, would be extremely beneficial.

7. PLOS authors have the option to publish the peer review history of their article (what does this mean? ). If published, this will include your full peer review and any attached files.

**Do you want your identity to be public for this peer review?** For information about this choice, including consent withdrawal, please see our Privacy Policy .

Reviewer #2: **Yes: ** Sepehr Khosravi

Reviewer #4: No

---

## [Author Response · Author response to Decision Letter 2]

27 Nov 2024

Bogotá, November 27th 2024

Dear Editor and reviewers of PLoS ONE

Once again, we thank you for your time and effort in reviewing our manuscript entitled "Predicting Diabetes Mellitus Metabolic Targets and Chronic Complications Transitions—Analysis Based on Natural Language Processing and Machine Learning Models.” Detailed responses to the reviewer’s comments are given below. In the manuscript, changes in content have been highlighted in red inside the main document.

Kind regards,

The authors

---

## [Decision Letter · Decision Letter 2]

20 Dec 2024

PONE-D-24-18044R2Predicting Diabetes Mellitus Metabolic Goals and Chronic Complications Transitions—Analysis Based on Natural Language Processing and Machine Learning Models.PLOS ONE

Dear Dr. Rodríguez-Lesmes,

Thank you for submitting your manuscript to PLOS ONE. After careful consideration, we feel that it has merit but does not fully meet PLOS ONE’s publication criteria as it currently stands. Therefore, we invite you to submit a revised version of the manuscript that addresses the points raised during the review process.

We look forward to receiving your revised manuscript.

Kind regards,

Ozra Tabatabaei-Malazy

Academic Editor

PLOS ONE

Journal Requirements:

Reviewers' comments:

Reviewer's Responses to Questions

**Comments to the Author**

1. If the authors have adequately addressed your comments raised in a previous round of review and you feel that this manuscript is now acceptable for publication, you may indicate that here to bypass the “Comments to the Author” section, enter your conflict of interest statement in the “Confidential to Editor” section, and submit your "Accept" recommendation.

Reviewer #4: All comments have been addressed

2. Is the manuscript technically sound, and do the data support the conclusions?

Reviewer #4: Yes

3. Has the statistical analysis been performed appropriately and rigorously? 

Reviewer #4: Yes

4. Have the authors made all data underlying the findings in their manuscript fully available?

Reviewer #4: Yes

5. Is the manuscript presented in an intelligible fashion and written in standard English?

Reviewer #4: Yes

6. Review Comments to the Author

Reviewer #4: 1. Employ Additional Visual Representations:

Include one or two essential graphs illustrating machine learning performance, such as: A confusion matrix for the optimal model to illustrate prediction accuracy for each class. A bar chart illustrating the F1-scores of various models to facilitate comparisons for readers.

2. Emphasize Feature Significance: Incorporate a basic feature importance plot for a primary machine learning model (e.g., Random Forest or Logistic Regression) to illustrate which factors exerted the most significant influence on predictions.

3. Concisely Elucidate Model Selections: Incorporate a brief paragraph into the main text (citing Appendix 3) that elucidates the rationale for the selection of specific machine learning models, highlighting the interpretability of Logistic Regression and the robustness of Random Forest.

4. Elucidate Data Preprocessing:

Concise preprocessing measures, including as oversampling and feature selection, were implemented to ensure the models adeptly managed unbalanced input.

5. Concentrate on Principal Outcomes: Identify a significant discovery from the machine learning models, such as the primary predictor for complications or metabolic objectives, and its correlation with clinical observations.

6. Examine Constraints Gently:

One restriction is the problem of imbalanced data or dependence on the quality of EHR text, which was partially mitigated using approaches such as NLP and oversampling.

7. PLOS authors have the option to publish the peer review history of their article (what does this mean? ). If published, this will include your full peer review and any attached files.

**Do you want your identity to be public for this peer review?** For information about this choice, including consent withdrawal, please see our Privacy Policy .

Reviewer #4: No

---

## [Author Response · Author response to Decision Letter 3]

2 Feb 2025

Please find the detailed responses as a document in the submission

---

## [Decision Letter · Decision Letter 3]

23 Feb 2025

PONE-D-24-18044R3Predicting Diabetes Mellitus Metabolic Goals and Chronic Complications Transitions—Analysis Based on Natural Language Processing and Machine Learning Models.PLOS ONE

Dear Dr. Rodríguez-Lesmes,

Thank you for submitting your manuscript to PLOS ONE. After careful consideration, we feel that it has merit but does not fully meet PLOS ONE’s publication criteria as it currently stands. Therefore, we invite you to submit a revised version of the manuscript that addresses the points raised during the review process.

We look forward to receiving your revised manuscript.

Kind regards,

Ozra Tabatabaei-Malazy

Academic Editor

PLOS ONE

Journal Requirements:

Additional Editor Comments (if provided):

Reviewers' comments:

Reviewer's Responses to Questions

**Comments to the Author**

1. If the authors have adequately addressed your comments raised in a previous round of review and you feel that this manuscript is now acceptable for publication, you may indicate that here to bypass the “Comments to the Author” section, enter your conflict of interest statement in the “Confidential to Editor” section, and submit your "Accept" recommendation.

Reviewer #4: All comments have been addressed

2. Is the manuscript technically sound, and do the data support the conclusions?

Reviewer #4: Yes

3. Has the statistical analysis been performed appropriately and rigorously? 

Reviewer #4: Yes

4. Have the authors made all data underlying the findings in their manuscript fully available?

Reviewer #4: Yes

5. Is the manuscript presented in an intelligible fashion and written in standard English?

Reviewer #4: Yes

6. Review Comments to the Author

Reviewer #4: It is evident that all comments have been addressed, and the paper exhibits a profound and proficient writing style. However, it is recommended that additional tables from the support document, entitled "S3 ML Methods Detail," such as Tables A3.1 and 3.2, be incorporated into the original body of the paper to provide further substantiation and clarity.

7. PLOS authors have the option to publish the peer review history of their article (what does this mean? ). If published, this will include your full peer review and any attached files.

**Do you want your identity to be public for this peer review?** For information about this choice, including consent withdrawal, please see our Privacy Policy .

Reviewer #4: No

---

## [Author Response · Author response to Decision Letter 4]

28 Feb 2025

Following the reviewer's recommendation, we have incorporated the two tables from “S3 ML Methods Details” into the main document (now Tables 7 and 8) and relocated their corresponding explanations to the results section.

---

## [Editor Report · Decision Letter 4]

4 Mar 2025

Predicting Diabetes Mellitus Metabolic Goals and Chronic Complications Transitions—Analysis Based on Natural Language Processing and Machine Learning Models.

PONE-D-24-18044R4

Dear Dr. Rodríguez-Lesmes,

We’re pleased to inform you that your manuscript has been judged scientifically suitable for publication and will be formally accepted for publication once it meets all outstanding technical requirements.

Kind regards,

Ozra Tabatabaei-Malazy

Academic Editor

PLOS ONE
---

## [Editor Report · Acceptance letter]

PONE-D-24-18044R4

PLOS ONE

Dear Dr. Rodríguez-Lesmes,

I'm pleased to inform you that your manuscript has been deemed suitable for publication in PLOS ONE. Congratulations! Your manuscript is now being handed over to our production team.

Kind regards,

on behalf of

Dr. Ozra Tabatabaei-Malazy

Academic Editor

PLOS ONE